**Data Availability Statement:** Data cannot be shared publicly because of legal restrictions on sharing a de-identified data set. Data are available

# Screening and treatment of familial hypercholesterolemia in a French sample of ambulatory care patients: A retrospective longitudinal cohort study

**Jean Ferrières**[1]*, **Victoria Banks**[2], **Demetris Pillas**[2], **Francesco Giorgianni**[2], **Laurene Gantzer**[3], **Beranger Lekens**[3], **Lea Ricci**[4], **Margaux Dova-Boivin**[4], **Jean-Vannak Chauny**[4], **Guillermo Villa**[5], **Gaelle Désaméricq**[4]

1 Department of Cardiology and UMR INSERM 1295, Toulouse Rangueil University Hospital, Toulouse, France, 2 Amgen (UK) Ltd, Uxbridge, United Kingdom, 3 GERS SAS, Boulogne-Billancourt, France, 4 Amgen (France) SAS, Boulogne-Billancourt, France, 5 Amgen (Europe) GmbH, Rotkreuz, Switzerland

* jean.ferrieres@univ-tlse3.fr

## Abstract

### Background and aims

Untreated Familial Hypercholesterolemia (FH) leads to premature morbidity and mortality. In France, its epidemiology and management are understudied in ambulatory care. We described the clinical profile, pharmacological management, and clinical outcomes in a French sample of FH patients.

### Methods

This was a retrospective longitudinal study on patients from The Health Improvement Network (THIN®) database in France, between October 2016-June 2019. Patients ≥18 years, with probable/definite FH based on the Dutch Lipid Clinic Network (DLCN) criteria were included. Baseline characteristics, lipid profile, lipid-lowering therapy (LLT), low-density lipoprotein-cholesterol (LDL-C) goal achievement; and disease management at 6-month of follow-up were analyzed.

### Results

116 patients with probable (n = 70)/definite (n = 46) FH were included (mean age:57.8±14.0 years; 56.0% women; 9.5% with personal history of cardiovascular events); 90 patients had data available at follow-up. At baseline, 77.6% of patients had LDL-C>190 mg/dL, 27.6% were not receiving LLTs, 37.9% received statins alone, 20.7% statins with other LLTs, and 7.7% other LLTs. High-intensity statins were prescribed to 11.2% of patients, 30.2% received moderate-intensity statins, and 8.6% low-intensity statins. Only 6.0% of patients achieved LDL-C goal. At 6-month of follow-up, statins discontinuation and switching were 22.7% and 2.3%, respectively. None of the patients received proprotein convertase subtilisin/kexin type 9 (PCSK9) inhibitors at baseline nor follow-up.

from the GERS SAS (contact via mail) for researchers who meet the criteria for access to confidential data. The data underlying the results presented in the study are available from VAN DER LAAN Chantal at chantal.vanderlaan@cegedim. com. There is no generic address. Of note, she is not an author of the manuscript and GERS Data is a Cegedim subsidiary. Even if some authors (LEKENS Beranger and GANTZER Laurene) belong to GERS, they are not responsible for data access.

**Funding:** This study was funded by Amgen.

**Competing interests:** I have read the journal's policy and the authors of this manuscript have the following competing interests: FJ received lecture fees from the companies Amgen, AstraZeneca, MSD, Sanofi and Servier. LG and BL have no conflict of interest to declare. PD, GF, RL, DBM, CJV, VG and DG own stock in and are fulltime employee of Amgen. BV was a contract worker at Amgen. LG and BL are full-time employees of GERS SAS. This does not alter our adherence to PLOS ONE policies on sharing data and materials.

## Conclusions

Despite the existence of effective LLTs, FH patients are suboptimally-treated, do not achieve LDL-C goal, and exhibit worsened pharmacological management over time. Future studies with longer follow-up periods and assessment of factors affecting LDL-C management, including lifestyle and diet, are needed.

## Introduction

Familial Hypercholesterolemia (FH) is a group of inherited genetic defects resulting in lifelong elevated levels of low-density lipoprotein-cholesterol (LDL-C) [1–5]. FH is more common than previously believed [6, 7]. Worldwide, it affects 1 in 311 [6] to 313 [7] people of every race and ethnicity [1], making it one of the most common serious genetic disorders [5, 6]. In Europe, the prevalence of FH in the general population is estimated at 0.3% [6, 7]. In France, the most recent population surveys estimated the prevalence of FH at about 0.83% [8].

When left untreated, FH unequivocally put patients at increased risk for premature morbidity and mortality due to atherosclerotic cardiovascular disease (ASCVD) [1, 2, 5, 9–14]. The risk of cardiac disease among FH patients increases between 13 [2] and 22 folds [15] compared with the general population. Further, FH patients are predicted to have 3.9 times more lifetime cardiovascular (CV) events than non-FH patients with a similar risk profile [16].

Different guidelines and statements agree on the importance of early diagnosis and adequate treatment as cornerstones to prevent premature CVD and improve overall survival of patients [17–19]. However, FH remains substantially underdiagnosed and undertreated [2]. In European countries, diagnosis rates range between 6% in Spain and 71% in the Netherlands, and these numbers seem to be overestimated [2]. Until now, late FH diagnosis remains the norm across all world regions, including Europe [20]. Further, a large body of evidence reports massive under-treatment of subjects with FH, including an insufficient dose of Lipid-Lowering Therapy (LLT), and late introduction of the treatment, when severe atherosclerosis had already developed [21–23]. The Copenhagen General Population Study reports that only 48% of FH patients received statins [24].

A global call to action was initiated to recognize FH as a health priority, acknowledging that considerable gains to health are achievable through optimal diagnosis and treatment [3].

In France, there is no clear knowledge about FH, its epidemiology and management in-depth, as there is a lack of longitudinal information from large databases [8, 25]. In light of the scarcity of available data, we aimed to explore the clinical profile and pharmacological management in patients with FH using data from a large French sample in ambulatory care, with the ultimate goal of informing advancements in patient care and outcomes.

## Materials and methods

### Study design and source of data

This was a retrospective longitudinal study, based on a subset of patients from The Health Improvement Network (THIN$_{®}$) database in France. THIN$_{®}$ is a large real-world medico-economic European database, powered by GERS Data, a Cegedim Health Data Division. It comprises fully anonymized and non-extrapolated electronic medical records (EMR) collected at the physicians-level and coded using the International Classification of Diseases, 10$^{th}$ Revision (ICD-10) codes. THIN$^{®}$ database includes information on prescribed medicines per patient, and those reimbursed by the French National Insurance Healthcare Fund (CNAM).

Data were collected by 2,000 General Practitioners (GPs), 130 cardiologists, and 40 endocrinologists, receiving 5.5 million patients regularly in their community office.

The database obtained approval from the French National Data Protection Authority (CNIL) for data collection since 2002. As the study was a retrospective analysis using secondary anonymized patient data only, no additional ethical approval was needed.

Since 2016, physicians are alerted to LDL-C levels >190 mg/dL, suggesting a case of FH, and invited to complete the Dutch Lipid Clinic Network (DLCN) score. These criteria are validated to be used in daily practice to identify patients with FH who warrant aggressive treatment, their use is widespread across Europe [19, 26], and are recommended to establish the clinical diagnosis of FH by the European Atherosclerosis Society (EAS) [2, 19]. The DLCN is a composite score that predicts the likelihood of a patient having FH and allows the classification of patients into probability groups of FH, ranging from unlikely, to possible, probable, or definite FH diagnosis. It is based on a combination of five dimensions: 1) Family history, i.e. having a first-degree relative with known premature coronary heart disease (CHD), or a first-degree relative with known LDL-C >95th percentile by age and sex for country, or a first-degree relative with tendon xanthoma and/or corneal arcus, or a child <18 years with LDL-C >95th percentile by age and sex for country; 2) Clinical history corresponds to premature CHD or premature cerebral or peripheral vascular disease; 3) Physical examination confirming the presence of tendon xanthoma or corneal arcus in a person <45 years; 4) high LDL-C levels; and 5) Molecular genetic testing and the finding of a causative mutation, if available. Definite Unlikely FH is diagnosed with a score between 0 and 2 points, possible FH diagnosis can be made with a score between 3 and 5 points, probable FH diagnosis corresponds to a score between 6 and 8 points, and FH is diagnosed with a score higher than 8 points [2, 18].

## Study population

Patients ≥18 years, with probable FH (6–8 points) or definite FH (>8 points) on the DLCN score at index date were included. The inclusion period extended between 01 October 2016 and 30 June 2019.

## Study period

The index date was the date of the first recording of the DLCN score of the patient by the consented physician into the database. A two-year lookback period was used to collect baseline characteristics and treatments prescribed. However, the baseline assessment was based on the closest measurement prior to the index date. After the baseline assessment, the patient was followed-up for a period of 6 months.

## Outcome measures

Baseline characteristics included: demographic characteristics (age and gender); clinical characteristics/laboratory measurements (lipid panel including total-cholesterol, LDL-C and triglycerides levels); basic CV risk factors (history of familial dyslipidemia, body mass index (BMI) calculated as weight in kg/height in m$^2$, and comorbidities (diabetes defined as diagnosis of diabetes and/or prescription/reimbursement of at least 3 oral antidiabetic agents and/or insulin in the last 2 years; hypertension: defined as diagnosis of hypertension and/or blood pressure (BP) ≥140/90 mmHg; and CVD defined as diagnosis of at least one of the following: myocardial infarction, chronic ischemic heart disease, stable angina, unstable angina, ischemic stroke, carotid artery disease, transient ischemic attack, peripheral artery disease, abdominal aortic aneurysm); LLT defined as any single prescription of statins, ezetimibe, statin and ezetimibe in fixed or separate combinations, proprotein convertase subtilisin/kexin type 9 (PCSK9)

inhibitors, statin and PCSK9 inhibitors combinations, or other LLTs (nicotinic acid, fibrates, bile acid sequestrants); statins intensity, categorized into high, medium and low-intensity based on the American College of Cardiology and American Heart Association (ACC/AHA) guideline [27]; and LDL-C goal achievement defined as the number (%) of patients who achieved LDL-C goal (LDL-C <130 mg/dL in patients without CVD, and LDL-C <70 mg/dL in those with CVD) [14].

The outcomes consisted of disease management at 6-month of follow-up, categorized as LLT discontinuation (defined as any LLT prescription at index date but no LLT prescription at 6-month of follow-up), switching (defined as different LLT/combination of LLT prescribed during follow-up than the LLT prescribed at index date, i.e. drug or drugs at index date no longer used during follow-up), increase (defined as one or more new LLT started during follow-up with index drugs also still continued), decrease (defined as one or more LLT at index date no longer taken at follow-up, but one or more LLT at index date still continued), or no change (defined as no change in LLT drugs prescribed between index date and follow-up); and cumulative all-cause hospitalizations during the follow-up period.

### Statistical considerations

All data were collected and analyzed at baseline and at the end of the 6-month of follow-up. Descriptive analyses were performed for all baseline characteristics and outcomes of interest. Continuous variables were described using mean, standard deviation (SD), median, quartiles and minimum and maximum values. Categorical variables were described using percentage and 95% confidence interval for each category.

## Results

### Study population

The analysis included 45,487 patients with an LDL-C level >190 mg/dL. Of these, 1,098 subjects (2.4%) had a DLCN score completed between 01 October 2016 and 30 June 2019 (Fig 1). Patients with unlikely FH (n = 435) or possible FH (n = 547) DLCN scores were excluded. Therefore, at baseline, the study cohort included 116 patients with probable FH (n = 70) or definite FH (n = 46) DLCN scores; of whom ten (8.6%) had a genetic confirmation of their FH status. Physical stigmata of FH were detected in 19 patients (16.3%). Ninety patients (77.6%) had data available for the 6-month follow-up; of whom 58 patients (64.4%) had at least one contact with the physician during this period.

The baseline characteristics of the cohort are shown in Table 1. Mean age was 57.8±14.0 years, 56.0% of the patients were women, and 56.0% had a family history of dyslipidemia. Within this cohort, 11.2% were obese, and 34.5% had at least one comorbidity; the most common comorbidity was hypertension (28.4%), followed by CVD (9.5%).

Fig 2 details the lipid panel of the study participants at baseline. Overall, 77.6% of patients had an LDL-C level >190 mg/dL; 5.2% had an LDL-C level >325 mg/dL. Only 10.3% had an LDL-C level <160 mg/dL, and 6.9% had an LDL-C level between 160 and 190 mg/dL. Of the cohort, 82.8% had a total-cholesterol level ≥200 mg/dL; 37.0% had a triglycerides level ≥150 mg/dL. Finally, just 6.0% of the cohort achieved LDL-C treatment goal at baseline.

### Change of clinical management during the follow-up period

Therapeutic management of FH patients is shown in Table 2. At baseline, 27.6% of the patients were not receiving any LLTs. While only 37.9% received statin monotherapy, 6.0% received ezetimibe alone, 20.7% were prescribed statins in combination with other LLTs (including

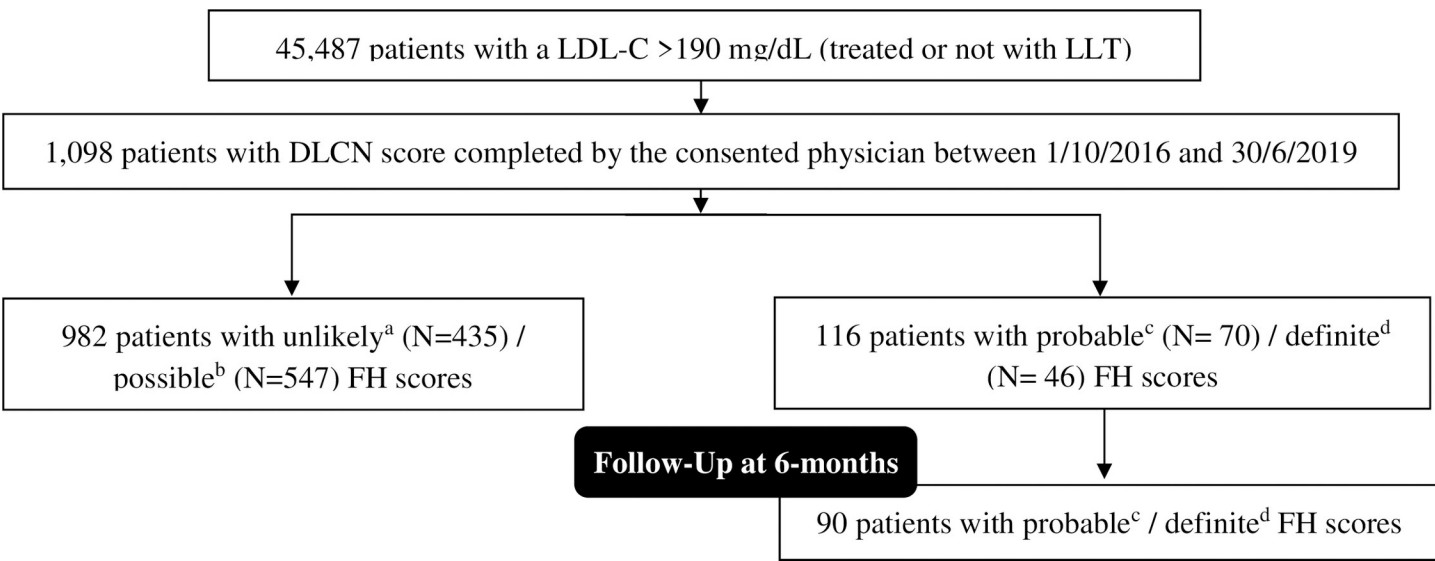

**Fig 1. Flowchart of study subjects.** LDL-C, Low-density lipoprotein cholesterol; LLT, Lipid-lowering therapy; DLCN, Dutch Lipid Clinic Network; FH, Familial hypercholesterolemia. [a]An unlikely FH refers to a DLCN score of 0 to 2 points. [b]A possible FH refers to a DLCN score of 3 to 5 points. [c]A probable FH refers to a DLCN score of 6 to 8 points. [d]A definite FH refers to a DLCN score of above 8 points.

15.5% with ezetimibe and 5.2% with LLTs other than ezetimibe). Finally, 7.7% of the patients received a combination of LLTs other than statins. Regarding statin intensity, only 11.2% of the study subjects were treated with high-intensity statins, 30.2% received moderate-intensity statins, and 8.6% were prescribed low-intensity statins (statin intensity was unknown for 8.6% of patients). At 6-month of follow-up, the percentage of patients not receiving any LLT increased to 47.8%; the percentage of patients receiving statins only decreased to 27.8%, and the percentage of patients prescribed statins and ezetimibe decreased to 12.2%. Further, the percentage of patients treated with high-intensity statins decreased to 8.9%, and those prescribed moderate-intensity statins dropped from to 20.0%. None of the patients were prescribed PCSK9 inhibitors at baseline nor at 6-month of follow-up.

Change in LLT prescription between baseline and follow-up is detailed in Fig 3, and S1 Table. The majority of patients (71.9%) without an LLT prescription at baseline maintained this status at follow-up. Around half of the patients (47.7%) receiving statins at baseline maintained this prescription at 6-month of follow-up, while 22.7% discontinued therapy, and only one patient (2.3%) was prescribed additional ezetimibe. More than half of the patients (55.6%) prescribed statins and ezetimibe at baseline maintained their status, while 5.6% stopped the treatment. Over time, statin users rarely switched therapies (2.3%), compared with those receiving a combination of statins and ezetimibe (11.1%) or other LLTs (14.3%).

More than half of the patients (53.8%) prescribed high-intensity statins at baseline maintained their status at 6-month of follow-up. Only 2.9% of those receiving moderate-intensity statins intensified their therapy at follow-up, while 42.9% maintained their baseline intensity. Ten percent of patients receiving low-intensity statins at baseline were prescribed a higher intensity at follow-up, while 30% kept their baseline intensity (S2 Table). During the follow-up period, only 3 patients (3.3%) were hospitalized.

## Discussion

To our knowledge, this is the first study in France to screen for FH in a cohort using ambulatory care EMR from a large database and to support GPs, and ambulatory cardiologists and

**Table 1. Demographic characteristics and comorbidities of patients with probable or definite FH at baseline.**

| | All patients | Probable FH | Definite FH |
|---|---|---|---|
| | N = 116, % [95% CI] | N = 70, % [95% CI] | N = 46, % [95% CI] |
| **Demographic Characteristics** | | | |
| Age, years, mean (SD) | 57.8 (14.0) | 60.9 (12.7) | 53.0 (14.7) |
| Gender | | | |
| Females | 65, 56.0 [46.5–65.2] | 40, 57.1 [44.7–68.9] | 25, 54.3 [39.0–69.1] |
| Males | 51, 44.0 [34.8–53.5] | 30, 42.9 [31.1–55.3] | 21, 45.7 [30.9–61.0] |
| **History of familial dyslipidaemia*** | 65, 56.0 [46.5–65.2] | 35, 50.0 [37.8–62.2] | 30, 65.2 [49.8–78.6] |
| **Comorbidities** | | | |
| Diabetes | 10, 8.6 [4.2–15.3] | 7, 10.0 [4.1–19.5] | 3, 6.5 [1.4–17.9] |
| Hypertension | 33, 28.4 [20.5–37.6] | 24, 34.3 [23.3–46.6] | 9, 19.6 [9.4–33.9] |
| CVD | 11, 9.5 [4.8–16.3] | 9, 12.9 [6.1–23.0] | 2, 4.3 [0.5–14.8] |
| Myocardial Infarction | - | - | - |
| Ischaemic Stroke | - | - | - |
| Peripheral Artery Disease | 2, 1.7 [0.2–6.1] | 1, 1.4 [<0.1–7.7] | 1, 2.2 [0.1–11.5] |
| CIHD | 4, 3.4 [0.9–8.6] | 2, 2.9 [0.3–9.9] | 2, 4.3 [0.5–14.8] |
| Stable Angina | 3, 2.6 [0.5–7.4] | 3, 4.3 [0.9–12.0] | - |
| Unstable Angina | - | - | - |
| Carotid Artery Disease | 2, 1.7 [0.2–6.1] | 2, 2.9 [0.3–9.9] | - |
| Transient Ischemic Attack | 2, 1.7 [0.2–6.1] | 2, 2.9 [0.3–9.9] | - |
| Abdominal Aortic Aneurysm | 1, 0.9 [<0.1–4.7] | 1, 1.4 [<0.1–7.7] | - |
| Obesity | | | |
| Non-obese | 71, 61.2 [51.7–70.1] | 39, 55.7 [43.3–67.6] | 32, 69.6 [54.2–82.3] |
| Obese | 13, 11.2 [6.1–18.4] | 7, 10.0 [4.1–19.5] | 6, 13.0 [4.9–26.3] |
| Missing | 32, 27.6 [19.7–36.7] | 24, 34.3 [23.3–46.6] | 8, 17.4 [7.8–31.4] |
| Any of the comorbidities listed above | 40, 34.5 [25.9–43.9] | 28, 40.0 [28.5–52.4] | 12, 26.1 [14.3–41.1] |

FH, Familial hypercholesterolemia; SD, Standard deviation; CI, Confidence interval; CVD, Cardiovascular disease; CIHD, Chronic ischemic heart disease.

*ICD-10 code of FH (E78.01).

endocrinologists in identifying those who need to be treated. Our results provide evidence on clinical profiles, pharmacological management, and LDL-C goal attainment in FH, thus informing care of this understudied patient-population in France.

## Clinical profiles

Underdiagnosis of FH patients remains a universal finding [28, 29], and is assumed to be more frequent among young adults, as they are less likely to refer to their physicians [30]. Recently, numerous efforts have been made in France to improve the national diagnosis and management of FH [31, 32]. Our study specifically pinpoints this issue in France, as reported by other large studies [8, 25], and adds evidence about the important role of primary care in identifying patients with FH, as found elsewhere [33–35]. Although being relatively young in age, patients included in our cohort study exhibit a cluster of CV risk factors, including poor LDL-C and triglycerides profiles at baseline, and a high level of comorbidities, all of which are known to be independently associated with CVD [36, 37], adding to the inherent CVD risk of FH [16], whether treated or not [38]; which was also shown in French patients [8]. It is worthy to note that in this study patients having a probable or definite FH were younger than those with unlikely or possible FH (mean age±SD: 57.8±14.0 vs. 61.5±12.6 years), and a greater proportion of them had LDL-C levels >190 mg/dL (77.6% vs. 67.0%) (S3 Table).

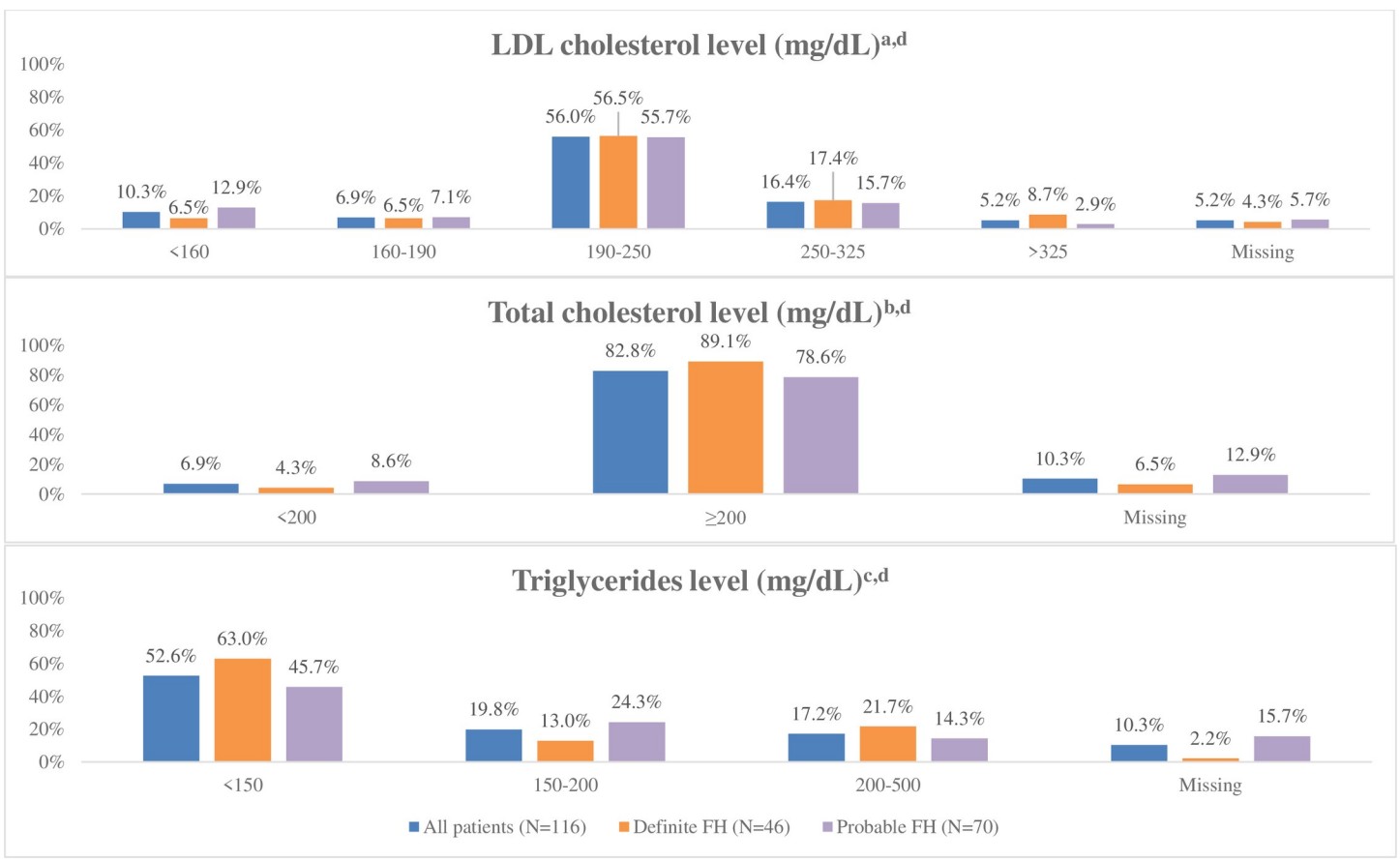

**Fig 2. Lipid panel of study subjects at baseline.** Min, Minimum; Max, Maximum; Q, Quartile. [a]Number of days to index-date: For all patients: Min: 0; Q1: 5; Q2: 43; Q3: 287; Max: 726; For patients with Definite FH: Min: 0; Q1: 8.5; Q2: 51.5; Q3: 333.5; Max: 687; For patients with Probable FH: Min: 0; Q1: 5; Q2: 25; Q3: 229; Max: 726. [b]Number of days to index-date: For all patients: Min: 0; Q1: 5.5; Q2: 43; Q3: 265.5; Max: 726; For patients with Definite FH: Min: 0; Q1: 15; Q2: 54; Q3: 347; Max: 687; For patients with Probable FH: Min: 0; Q1: 5; Q2: 22; Q3: 199; Max: 726. [c]Number of days to index-date: For all patients: Min: 0; Q1: 6; Q2: 47.5; Q3: 310.5; Max: 726; For patients with Definite FH: Min: 0; Q1: 15; Q2: 84; Q3: 347; Max: 709; For patients with Probable FH: Min: 0; Q1: 5; Q2: 25; Q3: 297; Max: 726. [d]The index date is the date of the first entry of the DLCN score of the patient by the consented physician into the database.

## Pharmacological management

Our findings confirm that, despite this elevated CV risk, FH in France is undertreated, and when LLT is provided, the therapy is suboptimal. LLT prescription in this cohort is not in line with the national guidelines for standard practice [14], as well as the European guidelines [19] that recommend prescribing first-line high-intensity statins in combination with diet and life-style changes among patients with an LDL-C level >190 mg/dL to reduce it by 50%, and consider adding PCSK9 inhibitors if failed to attain the goal [14]. Despite the high-quality evidence of the safety, effectiveness, and tolerance of PCSK9 inhibitors and even though these agents are reimbursed in France (although not in all forms of FH), none of the patients included in this study were prescribed PCSK9 inhibitors at baseline nor follow-up. Underuse of this therapy is a missed opportunity for improving patient outcomes; this underuse might be caused by numerous reasons. First, the initial prescription of these agents is reserved for specialties in cardiology, endocrinology, diabetes and metabolic diseases or internal medicine, while in our study, the patients were mostly seen by GPs. Second, the initial reimbursement of PCSK9 inhibitors in France was published between January and February 2018, while our data collection ended in June 2019; thus, most physicians could not prescribe. Third, the

**Table 2. Pharmacological management and clinical outcomes of patients with probable or definite FH, at baseline and at 6-month of follow-up.**

| | Baseline | Follow-up (month 6) |
|---|---|---|
| | N = 116, % [95% CI] | N = 90, % [95% CI] |
| **LLT prescription** | | |
| No LLT | 32, 27.6 [19.7–36.7] | 43, 47.8 [37.1–58.6] |
| Any LLT prescription | 84, 72.4 [63.3–80.3] | 47, 52.2 [41.4–62.9] |
| **Monotherapy** | | |
| Statins | 44, 37.9 [29.1–47.4] | 25, 27.8 [18.9–38.2] |
| Ezetimibe | 7, 6.0 [2.5–12.0] | 1, 1.1 [<0.1–6.0] |
| **Combined therapy** | | |
| Ezetimibe + Statins | 18, 15.5 [9.5–23.4] | 11, 12.2 [6.3–20.8] |
| Ezetimibe + Other LLT | 2, 1.7 [0.2–6.1] | 4, 4.4 [1.2–11.0] |
| Ezetimibe + Other LLT + Statins | 3, 2.6 [0.5–7.4] | - |
| Other LLT | 7, 6.0 [2.5–12.0] | 6, 6.7 [2.5–13.9] |
| Other LLT + Statins | 3, 2.6 [0.5–7.4] | - |
| **Statin intensity[a]** | | |
| Unknown | 10, 8.6 [4.2–15.3] | 7, 7.8 [3.2–15.4] |
| Low | 10, 8.6 [4.2–15.3] | 3, 3.3 [0.7–9.4] |
| Moderate | 35, 30.2 [22.0–39.4] | 18, 20.0 [12.3–29.8] |
| High | 13, 11.2 [6.1–18.4] | 8, 8.9 [3.9–16.8] |
| **Clinical Outcomes** | | |
| Hospitalizations, all causes | | |
| Number of patients hospitalized | - | 3, 3.3 [0.7–9.4] |

FH, Familial hypercholesterolemia; CI, Confidence interval; LLT, Lipid-lowering therapy; LDL-C, Low-density lipoprotein cholesterol.

[a]Any statin, excluding statin + ezetimibe fixed combination.

reimbursement of PCSK9 inhibitors was restricted to homozygous FH patients, and heterozygous FH patients treated with LCL-C apheresis, at the time the study was conducted (the reimbursement of PCSK9 inhibitors for secondary prevention of CV events, independently of FH status, was published in July 2020). Fourth, underuse of these agents might only be due to the short duration of follow-up (6 months), as PCSK9 inhibitors are used as second line agents, after failure of therapy with statins alone.

Our findings are in line with the results of the third French MONICA and MONALISA population surveys among patients with probable or definite FH and underscore the issue of undertreatment of FH patients in France [8]. Also, our results add to the international literature regarding the universal issue of undertreatment of FH patients, whereby treatment with LLTs ranged between 25.8% [39] in Canada to 48% in Denmark [24], 53% in the Netherlands [40] and 84% in Spain [41]. Hence, LDL-C goal achievement ranged between 3.4% in Spain [41] and 23% [40] in the Netherlands.

Besides highlighting the massive undertreatment of this population, our findings provide evidence on a worsened pharmacological management and underutilization of LLT over time, and these results are more alarming that what was reported elsewhere. For instance, data from the national screening program in the Netherlands [42] showed an increase in LLT prescription from 39% to 93% one-year following the initial examination. Besides, during the 6-month follow-up period, only 77.5% of the initial cohort were retained for evaluation, as follow-up data were not available for 22.5% of the cohort. Generally, numerous barriers on the part of

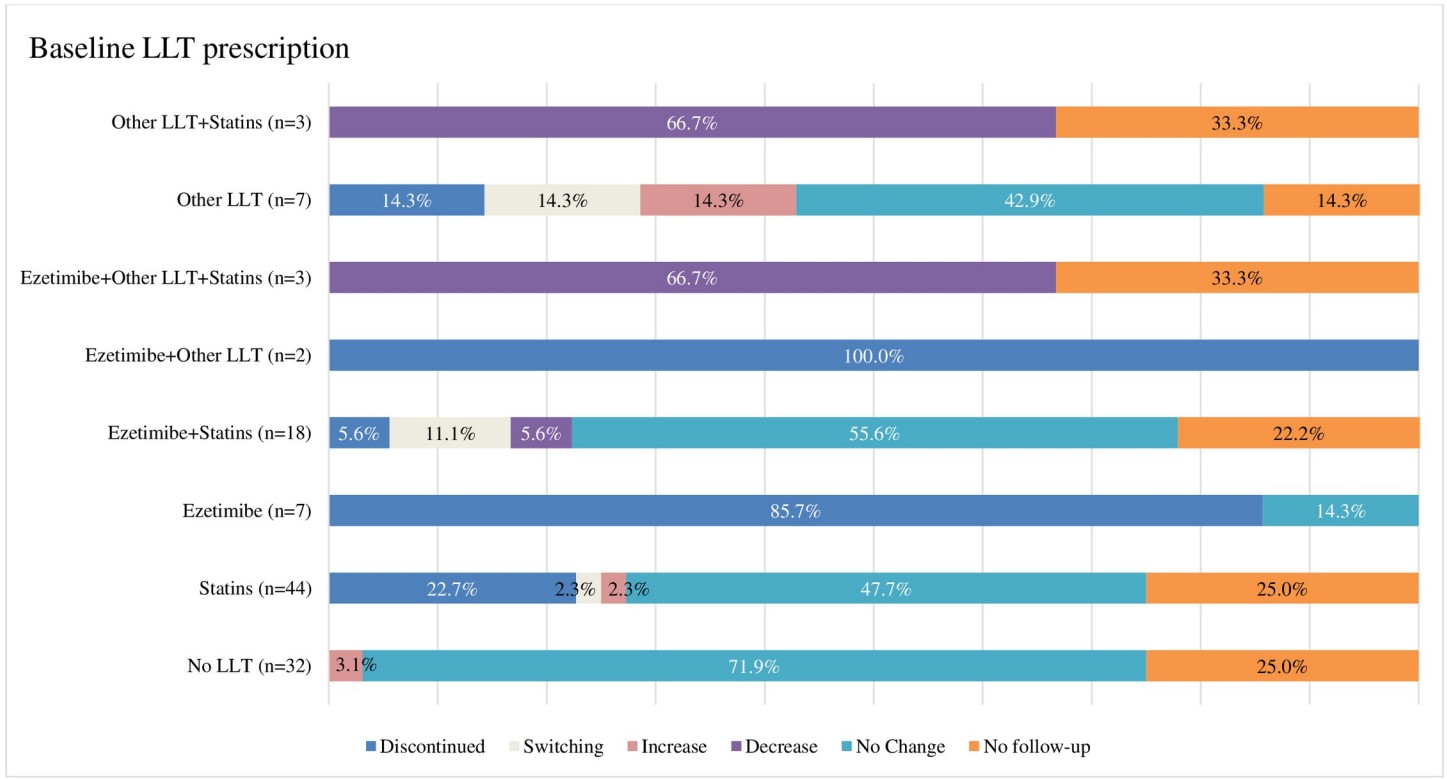

**Fig 3. LLT change status of patients with definite or probable FH at 6-month of follow-up.** Discontinued defined as any LLT prescription at index date but no LLT prescription at 6-month of follow-up. Switching defined as different LLT/combination of LLT prescribed during follow-up than the LLT prescribed at index date. Increase defined as one or more new LLT started during follow-up with index drugs also still continued. Decrease defined as one or more LLT at index date no longer taken at follow-up, but one or more LLT at index date still continued. No change defined as no change in LLT drugs prescribed between index date and follow-up.

patients, their families, physicians, and health systems hinder the treatment of the disorder [17]. Besides underdiagnosis at baseline which hinders treatment initiation, undertreatment might be due to the relatively young age of our cohort, and the occurrence of adverse reactions [43, 44], two factors that are strongly associated with poor compliance with LLTs. Also, this may be due to the suboptimal knowledge of FH and its management, as well as the underestimation of its CV risk by physicians providing care to FH patients in France, as shown by a recent survey in a representative sample of physicians [4]. Possibly, within the context of the "French Paradox", French physicians think that patients with FH have a lower coronary artery disease risk than FH patients elsewhere [8]. Likewise, young FH patients perceive their vulnerability to CVD as low [45], which may affect their adherence to LLTs [46]. As undertreatment of FH is associated with unfavorable outcomes, increasing compliance with LLTs should become a major concern for physicians. Early and frequent follow-up by physicians, especially lipid testing, is associated with improved adherence to LLTs [47].

## Strengths and limitations

This study uses real-world data from a national, large, established, and quality-assured database, where the information on FH was routinely and prospectively collected [48]. Using the DLCN criteria to diagnose FH is another strength of this study. Although genetic testing is the gold standard to ascertain the diagnosis of FH [14], recently, the Task Force for the management of dyslipidaemias of the European Society of Cardiology (ESC) and European

Atherosclerosis Society (EAS) recommend that FH should be diagnosed using clinical criteria and confirmed, when possible, via DNA analysis [19]. These criteria are most frequently used in population surveys in FH, as they show a good diagnostic and predictive value [26, 49, 50], and their use has been fostered among GPs [30]. Moreover, as the DLCN criteria were completed by the physician, the bias from missing values should be limited. Finally, we relied on prescribed and reimbursed medicines per patient.

On the other hand, based on the DLCN algorithm, a patient may obtain sufficient points to be categorized as a definite case solely from the non-LDL categories (family history, personal clinical history, physical exam). Besides, due to lack of time or by insufficient knowledge on FH and its risks by the physicians, selection bias may exist towards patients that are better followed-up by their physicians. Further, missing evaluation for some lifestyle factors limited the description of the characteristics of the patients included, and limited our ability to evaluate factors confounding the management of the patients. Additionally, there is the potential of misreporting of information in the existing health systems from where data are retrieved (EMR) as data are collected voluntarily. Moreover, familial combined hyperlipidemia or secondary causes of FH were not explored; the study did not entail molecular genetic testing to confirm FH status; hence all cases of probable/definite FH based on the DLCN score, irrespective of the cause, were included. This is a limitation of analyses using secondary patient data. Further, just as with all clinical database analyses, our findings dependent on the quality of information entered into the database. In some cases, incomplete or improper recording of some variables or diagnoses might take place, potentially leading to misclassification of some patients into subgroups. Due to the short-term duration of the study, we relied on surrogate outcomes and did not collect data on hard outcomes including mortality, and incidence recurrent CV events during follow-up (only all-cause hospitalizations during follow-up were reported). Also, data on LDL-C levels during follow-up were unavailable for the majority of patients, hindering our ability to assess attainment of treatment goal at follow-up. Besides, the short duration of the follow-up does not allow us to get the whole picture on second line treatment. Finally, our results may not be representative of the whole FH population in France, thus not be generalizable to this population.

## Conclusion

To our knowledge, this is the first study to explore clinical profiles and pharmacological management of clinically-diagnosed FH patients, as specified in international guidelines, in primary care practice in France. This study provides evidence that the diagnosis of FH is possible within general medicine, and integral to the GPs' activity; the latter are in the best position to identify patients before they manifest CV events. Also, our findings provide real-world evidence that FH patients in France are underdiagnosed, undertreated, and do not attain therapeutic LDL-C goals in ambulatory care settings, although effective therapy is available. Because of the debilitating clinical, societal and economic burden of untreated FH, it is imperative to escalate screening, and diagnosing of FH, intensify LLTs and make use of novel options such as PCSK9 inhibitors to reduce patients' CV risk, improve their prognoses, and alleviate the disease's burden.

## Supporting information

**S1 Table. LLT prescriptions of patients with definite or probable FH, at baseline and at month-6 of follow-up[a].** LLT, Lipid-lowering therapy; FH, Familial hypercholesterolemia.
[a]Data are presented as n (%).
(DOCX)

**S2 Table. Statin intensity of patients with definite or probable FH, at baseline and at month-6 of follow-up[a].** [a]Data are presented as n (%). [b]Any statin, excluding statin + ezetimibe fixed combination. FH, Familial hypercholesterolemia.
(DOCX)

**S3 Table. Demographic and clinical characteristics and comorbidities of patients with definite, probable, possible or unlikely FH, at baseline[a].** FH, Familial hypercholesterolemia; SD, Standard deviation; CI, Confidence interval; CVD, Cardiovascular disease; CIHD, Chronic ischemic heart disease; LDL, Low-density lipoprotein; Min, Minimum; Max, Maximum; Q, Quartile; LLT, Lipid lowering therapy. [a]The index date is the date of the first entry of the DLCN score of the patient by the consented physician into the database. [b]Any statin, excluding statin + ezetimibe fixed combination.
(DOCX)

## Acknowledgments

Medical writing assistance was provided by Nadine Saleh (PharmD, PhD) of ESN for research assistance.

## Author Contributions

**Conceptualization:** Margaux Dova-Boivin, Jean-Vannak Chauny, Gaelle Désaméricq.

**Data curation:** Demetris Pillas, Francesco Giorgianni, Laurene Gantzer, Beranger Lekens.

**Formal analysis:** Victoria Banks, Demetris Pillas, Francesco Giorgianni.

**Funding acquisition:** Margaux Dova-Boivin, Jean-Vannak Chauny, Gaelle Désaméricq.

**Methodology:** Margaux Dova-Boivin, Jean-Vannak Chauny, Gaelle Désaméricq.

**Project administration:** Gaelle Désaméricq.

**Supervision:** Gaelle Désaméricq.

**Writing – review & editing:** Jean Ferrières, Victoria Banks, Demetris Pillas, Francesco Giorgianni, Laurene Gantzer, Beranger Lekens, Lea Ricci, Margaux Dova-Boivin, Jean-Vannak Chauny, Guillermo Villa, Gaelle Désaméricq.

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
