## [Decision Letter · Decision Letter 0]

23 Mar 2021

PONE-D-20-38253

Screening and treatment of Familial Hypercholesterolemia in a French sample of ambulatory care patients: a retrospective longitudinal cohort study

PLOS ONE

Dear Dr. Ferrières,

Thank you for submitting your manuscript to PLOS ONE. After careful consideration, we feel that it has merit but does not fully meet PLOS ONE’s publication criteria as it currently stands. Therefore, we invite you to submit a revised version of the manuscript that addresses the points raised during the review process.

We look forward to receiving your revised manuscript.

Kind regards,

Antonio Palazón-Bru, PhD

Academic Editor

PLOS ONE

Journal Requirements:

4. Thank you for stating the following in the Financial Disclosure section:

"This study was funded by Amgen."

We note that one or more of the authors have an affiliation to the commercial funders of this research study : Amgen Ltd, Amgen SAS, Amgen (Europe) GmbH.

Thank you for stating the following in the Competing Interests section:

"I have read the journal's policy and the authors of this manuscript have the following competing interests: FJ received lecture fees from the companies Amgen, AstraZeneca, MSD, Sanofi and Servier. LG and BL have no conflict of interest to declare. PD, GF, RL, DBM, CJV, VG and DG own stock in and are full-time employee of Amgen. BV was a contract worker at Amgen."

We note that one or more of the authors are employed by a commercial company: GERS SAS.

4.1. Please provide an amended Funding Statement declaring these commercial affiliations, as well as a statement regarding the Role of Funders in your study. If the funding organizations did not play a role in the study design, data collection and analysis, decision to publish, or preparation of the manuscript and only provided financial support in the form of authors' salaries and/or research materials, please review your statements relating to the author contributions, and ensure you have specifically and accurately indicated the role(s) that these authors had in your study. You can update author roles in the Author Contributions section of the online submission form.

Reviewers' comments:

Reviewer's Responses to Questions

**Comments to the Author**

1. Is the manuscript technically sound, and do the data support the conclusions?

Reviewer #1: Yes

Reviewer #2: Yes

2. Has the statistical analysis been performed appropriately and rigorously? 

Reviewer #1: Yes

Reviewer #2: Yes

3. Have the authors made all data underlying the findings in their manuscript fully available?

Reviewer #1: No

Reviewer #2: Yes

4. Is the manuscript presented in an intelligible fashion and written in standard English?

Reviewer #1: Yes

Reviewer #2: Yes

5. Review Comments to the Author

Reviewer #1: The manuscript has studied the follow up of patients with FH . The Manuscript has been written in a well designed , and would be acceptable for publication. I have the following question:

1- What is the TG cut of for evalauting FH?

2- how the authors were assure that these patients are not familial combined hyperlipidemia?

3- Did the authors conduct genetic study?

Reviewer #2: The authors present data on characteristics and treatment patterns of 116 patients with FH in France.

Overall the findings are consistent with many other studies in other jurisdictions, including in France. The study is sound. The sample size is very small. A number of issues could be improved:

1. More data on how DLCN scores were derived are needed. How often was genetic testing used? In what percentage of patients were physical stigmata of FH detected?

2. How were secondary caused of hypercholesterolemia excluded?

3. The most notable finding was that of the 45,487 patients with an LDL-C level >190 mg/dL for whom the physician was alerted to calculate a DLCN score, a score was calculated in only 1098 patients. More details about what accounts for this gap are required. Was the primary limitation lack of familiarity with the DLCN score (and what decision support was provided to clinicians?).

4. It would also be helpful to show how many patients had an ICD-10 code of FH (E78.01) and what was the overalp with the patients win whom a DLCN score was calculated.

5. The lack of use of PCSK9 inhibitors is notable and in contrast to other contemporary cohorts of patients with FH. Does this relate to reimbursement criteria in France?

6. Some of the references are innapropriate. For instance, on page 17: "whereby

336 treatment with LLTs ranged between 25.8% [40] in Canada". Ref 40 refers to a study in the Netherlands, not in Canada. Please provide appropriate references for these statements.

7. The authors seem to overestimate the impact of the work. Eg "This study pioneers in generating national data regarding the management of FH patients in France". In fact, other studies have previously reported on FH in France such as reference 8 in the manuscript. Suggest toning down some of these conclusions.

6. PLOS authors have the option to publish the peer review history of their article (what does this mean?). If published, this will include your full peer review and any attached files.

Reviewer #1: No

Reviewer #2: No

---

## [Author Response · Author response to Decision Letter 0]

14 Jun 2021

Dear editor, 

Thank you for your email dated March 23rd, 2021, enclosing the reviewers’ comments. We would like to thank the editorial board and the reviewers for the careful and thorough review of this manuscript, and for their thoughtful comments and constructive suggestions, which we feel have substantially contributed to improve our manuscript. 

We have carefully reviewed the comments and have revised the manuscript accordingly. We have addressed the comments that were raised by the reviewers and our responses are presented in a point-by-point manner below. 

The changes implemented are tracked in the manuscript. The revision has been developed in consultation with all coauthors, and each coauthor has given approval to the final version of the manuscript.

We hope that the responses are satisfactory and that the revised manuscript is now suitable for publication. We look forward to hearing from you soon.

Jean Ferrieres, MD, PhD, FESC

Corresponding author

Reviewer #1: The manuscript has studied the follow up of patients with FH. The Manuscript has been written in a well-designed, and would be acceptable for publication. I have the following question:

We thank the reviewer for the effort invested in reviewing this manuscript and for their interesting comments.

1- What is the TG cut off for evaluating FH?

Indeed, the classification of familial hypercholesterolemia (FH) in this study was based on the Dutch Lipid Clinic Network (DLCN) score, as per the European Atherosclerosis Society (EAS) guidelines [1]. This score relies on five criteria: family history, clinical history of premature coronary heart disease (CHD) or cerebral or peripheral vascular disease, physical examination for tendon xanthomas or corneal arcus, very high LDL cholesterol (LDL-C) level on repeated measurements, and/or a causative mutation detected by molecular genetic testing. Triglycerides (TG) level is not within these criteria; thus, it was not used to evaluate FH. TG levels are merely reported as part of the description of the study population [Figure 2 and S3 Table in the manuscript].

[1] Nordestgaard BG, Chapman MJ, Humphries SE, Ginsberg HN, Masana L, Descamps OS, Wiklund O, Hegele RA, Raal FJ, Defesche JC, Wiegman A. Familial hypercholesterolaemia is underdiagnosed and undertreated in the general population: guidance for clinicians to prevent coronary heart disease: consensus statement of the European Atherosclerosis Society. European heart journal. 2013 Dec 1;34(45):3478-90.

2- How the authors were assuring that these patients are not familial combined hyperlipidemia? 

In French hospitals and with specialist physicians, when a patient presents with a potential case of FH, after calculating the DLCN score and assess family tree, a measurement of apolipoprotein B (Apo B) is conducted. If the latter is shown to be high, this might indicate a case of familial combined hyperlipidemia. However, in an ambulatory setting and with general practitioners (GPs), as it is the case with this study, the measurement of Apo B is not done, since it is not reimbursed. Moreover, the confirmation of familial hyperlipidemia requires molecular genetic testing, which was not part of the information collected in the database used for this study. Accordingly, in this study, it cannot be ensured that patients do not have familial combined hyperlipidemia, since both measurement of Apo B and molecular genetic testing were not conducted nor available. This has been added to the limitations of the study [Lines 387-390 of the revised manuscript].

3- Did the authors conduct genetic study?

Only 10 patients (8.6%) had genetic confirmation of their FH status at the time of the DLCN assessment [Lines 185-186 in the revised manuscript]. These patients already underwent molecular genetic testing. However, molecular genetic testing was not an inclusion criterion in the current study, nor did the study entail molecular genetic testing to confirm FH status. This has been added to the limitations of the study [Lines 387-390 of the revised manuscript].

Reviewer #2: The authors present data on characteristics and treatment patterns of 116 patients with FH in France.

Overall the findings are consistent with many other studies in other jurisdictions, including in France. The study is sound. The sample size is very small. A number of issues could be improved:

We thank the reviewer for the thoughtful comments and recommendations, which were considered in the revised version of the manuscript. Specifically, all changes suggested by the reviewer were added in track changes in the revised version of the manuscript.

1. More data on how DLCN scores were derived are needed. How often was genetic testing used? In what percentage of patients were physical stigmata of FH detected?

The calculation of the DLCN score is done by the physician on a voluntary basis. During the medical visit, the physician received pop-up style windows from the electronic medical record (EMR) system if LCL-C>190 mg/dL, offering her/him to carry out screening for FH for the patient. The physician had to choose between 4 options: “yes”; “no, not now”; “no, don't ask me again for this patient” and “no, no longer ask me for all my patients”. If the physician selected “yes”, she/he were then invited to fill the DLCN score, thus providing additional or missing information not yet collected in the EMR for this specific patient.

Only 10 patients (8.6%) had genetic confirmation of their FH status at the time of the DLCN assessment [Lines 185-186 in the revised manuscript]. These patients already underwent molecular genetic testing. However, molecular genetic testing was not an inclusion criterion in the current study, nor did the study entail molecular genetic testing to confirm FH status. This has been added to the limitations of the study [Lines 387-390 of the revised manuscript].

Physical stigmata of FH were detected in 19 patients (16.3%) at the time of the DLCN assessment, this information has been added in the “Results / Study Population” section [Lines 186-187 of the revised manuscript].

2. How were secondary caused of hypercholesterolemia excluded?

Secondary causes of FH were not explored. Hence, all cases of FH, irrespective of the cause, were included. This has been added to the limitations of the study [Lines 387-390 of the revised manuscript].

3. The most notable finding was that of the 45,487 patients with an LDL-C level >190 mg/dL for whom the physician was alerted to calculate a DLCN score, a score was calculated in only 1098 patients. More details about what accounts for this gap are required. Was the primary limitation lack of familiarity with the DLCN score (and what decision support was provided to clinicians?).

The calculation of the DLCN score is done by the physician on a voluntary basis. During the medical visit, the physician received pop-up style windows from the EMR system if LCL-C>190 mg/dL, offering her/him to carry out screening for FH for the patient. The physician had to choose between 4 options: “yes”; “no, not now”; “no, don't ask me again for this patient” and “no, no longer ask me for all my patients”. If the physician selected “yes”, she/he were then invited to fill the DLCN score, thus providing additional or missing information not yet collected in the EMR for this specific patient. 

The reasons behind not calculating the DLCN score were not reported in the database used for this study, and were not part of the aims of the current study. Burdensome pop-up alerts and clinical inertia may remain.

4. It would also be helpful to show how many patients had an ICD-10 code of FH (E78.01) and what was the overalp with the patients win whom a DLCN score was calculated.

This information is available in Table 1 [History of familial FH: N=65, 56.0%; 95%CI: 46.5 - 65.2]. For clarity, a footnote has been added to Table 1 providing further information on the exact ICD-10 code of FH [Line 210 of the revised manuscript].

5. The lack of use of PCSK9 inhibitors is notable and in contrast to other contemporary cohorts of patients with FH. Does this relate to reimbursement criteria in France?

Several reasons behind the underuse of PCSK9 inhibitors were already discussed in the “Discussion / Pharmacological management” section in the manuscript [Lines 320-331 of the revised manuscript], including: 1) the initial prescription of these agents that is reserved for specialties in cardiology, endocrinology, diabetes and metabolic diseases or internal medicine, while in this study, the patients were mostly seen by GPs; 2) the initial reimbursement of PCSK9 inhibitors in France that was published between January and February 2018, while the data collection in this study ended in June 2019; thus, most GPs could not prescribe them; 3) the reimbursement of PCSK9 inhibitors was restricted only to homozygous FH patients and heterozygous FH patients treated with LCL-C apheresis, at the time the study was conducted; 4) PCSK9 inhibitors are used as second line agents, after failure of therapy with statins alone, which may not be captured given the short duration of follow-up (6 months) in this study.

6. Some of the references are innapropriate. For instance, on page 17: "whereby 336 treatment with LLTs ranged between 25.8% [40] in Canada". Ref 40 refers to a study in the Netherlands, not in Canada. Please provide appropriate references for these statements.

We thank the reviewer for pointing out this referencing error. The references have been corrected in the revised version of the manuscript.

7. The authors seem to overestimate the impact of the work. Eg "This study pioneers in generating national data regarding the management of FH patients in France". In fact, other studies have previously reported on FH in France such as reference 8 in the manuscript. Suggest toning down some of these conclusions.

This study on FH uses real-world data from a national, large, established, and quality-assured database, where the information on FH was routinely and prospectively collected. This is not the case of previous analyses of FH in France [1]. However, as per the reviewer’s comment, the sentence to describe this study has been changed as follows “This study uses real-world data from a national, large, established, and quality-assured database, where the information on FH was routinely and prospectively collected” [Line 365-368 of the revised manuscript].

[1] Bérard E, Bongard V, Haas B, Dallongeville J, Moitry M, Cottel D, Ruidavets JB, Ferrières J. Prevalence and treatment of familial hypercholesterolemia in France. Canadian Journal of Cardiology. 2019 Jun 1;35(6):744-52.

---

## [Decision Letter · Decision Letter 1]

15 Jul 2021

Screening and treatment of Familial Hypercholesterolemia in a French sample of ambulatory care patients: a retrospective longitudinal cohort study

PONE-D-20-38253R1

Dear Dr. Ferrières,

We’re pleased to inform you that your manuscript has been judged scientifically suitable for publication and will be formally accepted for publication once it meets all outstanding technical requirements.

Kind regards,

Antonio Palazón-Bru, PhD

Academic Editor

PLOS ONE

Additional Editor Comments (optional):

Reviewers' comments:

Reviewer's Responses to Questions

**Comments to the Author**

1. If the authors have adequately addressed your comments raised in a previous round of review and you feel that this manuscript is now acceptable for publication, you may indicate that here to bypass the “Comments to the Author” section, enter your conflict of interest statement in the “Confidential to Editor” section, and submit your "Accept" recommendation.

Reviewer #2: All comments have been addressed

2. Is the manuscript technically sound, and do the data support the conclusions?

Reviewer #2: Yes

3. Has the statistical analysis been performed appropriately and rigorously? 

Reviewer #2: Yes

4. Have the authors made all data underlying the findings in their manuscript fully available?

Reviewer #2: No

5. Is the manuscript presented in an intelligible fashion and written in standard English?

Reviewer #2: Yes

6. Review Comments to the Author

Reviewer #2: (No Response)

7. PLOS authors have the option to publish the peer review history of their article (what does this mean?). If published, this will include your full peer review and any attached files.

Reviewer #2: No

---

## [Editor Report · Acceptance letter]

23 Jul 2021

PONE-D-20-38253R1 

Screening and treatment of Familial Hypercholesterolemia in a French sample of ambulatory care patients: a retrospective longitudinal cohort study. 

Dear Dr. Ferrières:

I'm pleased to inform you that your manuscript has been deemed suitable for publication in PLOS ONE. Congratulations! Your manuscript is now with our production department. 

Kind regards, 

on behalf of

Dr. Antonio Palazón-Bru 

Academic Editor

PLOS ONE